# Health Consequences for E-Waste Workers and Bystanders—A Comparative Cross-Sectional Study

**DOI:** 10.3390/ijerph17051534

**Published:** 2020-02-27

**Authors:** Damian Fischer, Fatima Seidu, Jennie Yang, Michael K. Felten, Cyryl Garus, Thomas Kraus, Julius N. Fobil, Andrea Kaifie

**Affiliations:** 1Institute for Occupational, Social, and Environmental Medicine, Medical Faculty, RWTH Aachen University, Pauwelsstrasse 30, 52074 Aachen, Germany; damian.fischer@rwth-aachen.de (D.F.); jennie.yang@rwth-aachen.de (J.Y.); mfelten@ukaachen.de (M.K.F.); cgarus@ukaachen.de (C.G.); tkraus@ukaachen.de (T.K.); 2GIZ Ghana Country Office, 7 Volta Street, Accra, Ghana; seidufa@yahoo.de; 3Department of Biological, Environmental and Occupational Health Sciences, School of Public Health, University of Ghana, P.O. Box LG13, Legon, Ghana; jfobil@ug.edu.gh

**Keywords:** occupational exposure, occupational health and safety, Agbogbloshie, ergonomic burden, back pain, drug abuse

## Abstract

Informal e-waste recycling is associated with several health hazards. Thus far, the main focus of research in the e-waste sector has been to assess the exposure site, such as the burden of heavy metals or organic pollutants. The aim of this study was to comprehensively assess the health consequences associated with informal e-waste recycling. A questionnaire-based assessment regarding occupational information, medical history, and current symptoms and complaints was carried out with a group of *n* = 84 e-waste workers and compared to a control cohort of *n* = 94 bystanders at the e-waste recycling site Agbogbloshie. E-waste workers suffered significantly more from work-related injuries, back pain, and red itchy eyes in comparison to the control group. In addition, regular drug use was more common in e-waste workers (25% vs. 6.4%). Both groups showed a noticeable high use of pain killers (all workers 79%). The higher frequency of symptoms in the e-waste group can be explained by the specific recycling tasks, such as burning or dismantling. However, the report also indicates that adverse health effects apply frequently to the control group. Occupational safety trainings and the provision of personal protection equipment are needed for all workers.

## 1. Introduction

Year after year, finding a solution to the global problem of electric and electronic waste (e-waste) is becoming increasingly urgent [1]. In West Africa, Ghana and Nigeria, as the main import centers, have established themselves as the most important trade routes for used electrical and electronic equipment (UEEE) on the continent [2]. UEEE at the end of its life span, especially that imported from Europe, leads to the creation of hundreds of thousands of tons of electronic waste every year in Ghana alone (179,000 t in 2009) [2].

The accumulation of this amount of electronic waste drives the growth of the second largest e-waste processing site in West Africa, which has achieved international attention as one of the world’s Top 10 toxic threats 2013 (according to Pure Earth/Blacksmith Institute) [3]. Under minimal and inefficient governmental control, an informal sector of e-waste recycling emerged west of the capital of Ghana, Accra, more than two decades ago [4]. The 16 km^2^ Agbogbloshie district in the Korle Lagoon, known locally as Sodom and Gomorrah, was leased in 1994 by the informal Greater Accra Scrap Dealers’ Association of Ghana (GASDA) from the National Youth Authority (NYA) [4,5]. 

Job opportunities on the scrap yard attract migrants from the disadvantaged rural areas in the north of the country, particularly for members of ethnic minority groups whose chances of finding a job in the formal sector are virtually nonexistent [2]. In 2016, 3000 workers were employed at the scrap yard, of whom 1500 were registered with GASDA [4]. The primitive methods used to recover valuable metals from the hazardous electronic waste expose workers to high levels of occupational risks. The most basic tools are used to dismantle the devices. Cables are burned using insulating foam from dismantled refrigerators or car tires as fuel to melt off their insulation. Most of the workers are not wearing any personal protection against the highly contaminated smoke and suffer from cuts and burns [6].

The burden on the workers was clearly shown in previous research: soil samples from Agbogbloshie showed significantly elevated levels of trace metals and PBDE (polybrominated diphenyl ethers) [7,8,9]. In personal air samples, increased concentrations of aluminum, copper, iron, lead, and zinc values could be detected [7]. Cadmium and lead were significantly elevated in blood samples of e-waste workers [10,11]. Urine samples also showed elevated levels of PAH (polycyclic aromatic hydrocarbons) [12], cadmium, chromium, nickel [11], iron, antimony, lead, and various arsenic species [10,13].

However, the most acute medical needs of the scrapyard workers remain unassessed. Most published studies focus on the external exposure to hazardous substances, but not the health consequences caused by this exposure. Health effects associated with e-waste processing, in particular in vulnerable groups, such as children and pregnant women, have been partially described previously but still remain neglected [14]. In interviews with e-waste workers carried out by Asampong et al. (2015), serious urgent health problems at the scrapyard were addressed for the first time [15].

Therefore, it seems obvious that a comprehensive analysis, not only of the exposure side (stresses) but also of the health consequences caused by the exposure (strains), is essential for an efficient occupational medical intervention for e-waste workers in order to minimize occupational hazards and improve their health and safety. Regarding the assessment of health effects associated with e-waste processing in Agbogbloshie, only two studies addressed the occupation related stress–strain axis. Burns et al. interviewed workers and found an association between noise exposure and the elevation in average heart rate as well as the number of injuries [16,17]. In Nigeria, the other important West African UEEE hub, only one study investigated a high rate of injuries among workers in the e-waste sector [18].

The focus of this study was to comprehensively assess the health effects of e-waste processing at the Agbogbloshie recycling site. To differentiate the additional health burden that is associated with e-waste recycling, we compared our findings in e-waste workers with those working in the vicinity of the scrapyard and frequent visitors not actively involved in recycling work (bystanders).

## 2. Materials and Methods

### 2.1. Study Population

The study was carried out in May 2019 at Agbogbloshie, Accra, Ghana. Subjects were recruited at a medical care unit at the recycling site as part of a free two-day health checkup by the Ghana Health Service and directly at their place of work during visits over three weeks. This study was approved by the Ethics Committee of Rheinisch-Westfälische Technische Hochschule Aachen University (EK 083-19) and all subjects gave their informed consent for participation. The study population consisted of e-waste workers (*n* = 84) and non-e-waste worker (bystander, *n* = 94). The e-waste workers were allocated to occupational subgroups according to the predominant job tasks of dismantlers (*n* = 52), burners (*n* = 21), and collectors (*n* = 11). The dismantlers dismantle the electrical and electronic devices, the burners burn cables and other components, and the collectors search the ashes for valuable metals. The control group consisted of people working in and around the Agbogbloshie recycling site without being directly involved into e-waste processing. Among the non-e-waste workers (bystander), onion carriers (*n* = 28) from the adjacent onion market, scraps traders (*n* = 15), sellers of food and other articles (*n* = 33), metal workers (*n* = 6), repairers (*n* = 3), and others (e.g., tailors, security guards, and drivers; *n* = 9) were identified. Onion carriers, for example, loaded onion sacks from and onto trucks by the main road, scraps traders provided workers with devices, and sellers crossed the scrap yard daily with their goods coming from shops at the periphery of the yard.

### 2.2. Questionnaire

All subjects completed a questionnaire in the presence of the study team, which was supported by an interpreter if required. The questionnaire was completed at the medical care unit during the health checkup or directly at the workplace, in dependence of the recruitment situation. The questionnaire included items on demographic characteristics, the occupational situation, and medical history. The questionnaire was divided into the following five sections:Personal information: Age, sex, marital status, level of education, religion, and residency.Occupational information: Working with e-waste, specific task, duration of working, use of personal protection equipment, and specification of control group occupation.Habits/Lifestyle: Dietary habits, smoking, and drug use.Medical information: Current complaints, such as diseases of the skin, infections, psychiatric disorders, diseases of the eyes and ears, cardiovascular diseases, lung diseases, musculoskeletal diseases, and injuries.Pre-existing medical care: Location of medical care, frequency, and health insurance status.

Work exposure was queried with items on skin contact with chemicals or metals; volume exposure; inhalation of smoke, dust, or gases; great physical stress at work; and traffic accidents during work. Concerning skin symptoms and shortness of breath, the subjects were specifically asked whether they suspected an association with their work. Given the lack of clinical diagnosis, the questions were prepared in accordance with ICD-10 (International Statistical Classification of Diseases and Related Health Problems) specifications and after thorough consultation with occupational physicians and local project support.

### 2.3. Statistical Analyses

Clinical data were collected and analyzed using SAS Software (SAS 7.1, SAS Institute Inc., Cary, NC, USA). First, descriptive analyses of general characteristics, medical conditions, and medical care and insurance status were performed for characterization of the cohort. Possible associations between current symptoms, diseases, and specific work tasks were investigated. We used chi-square tests to describe the distribution of categorical variables between the different subtypes. All statistical tests were two-sided, and *p* < 0.05 was used as the level of significance.

## 3. Results

The study population (*n* = 178) consisted of *n* = 84 e-waste workers (EW) and *n* = 94 bystanders (BY). Figure 1 shows the distribution of the EW and BY into their respective occupational subgroups.

There were no age differences between the groups (EW mean age = 27 years, range = 18–59 years; BY mean age = 28 years, range = 18–50 years) (Table 1). Among the EW, 97.6% were male, while, among the BY, 71.3% were male. School education differed between the groups with more subjects without school education for BY (EW 26.2% and BY 43.0%) (Table 1). The origin of the EW and BY differed significantly, as shown in Table 1 (100% of EW were from Ghana and 59.6% or BY were from Ghana).

As shown in Table 2, significant differences between the two groups were found in substance abuse, such as the inhalation of cannabis (EW 25.0% and BY 6.4%). The use of a regular medication, mainly in the form of painkillers, was elevated among both groups (57.5%). Regarding access to medical care, the subjects were asked about registration in the national health insurance, as well as the use of different institutions to access medical care (Table 2). Most subjects got access to medical care by visiting pharmacies (EW 82.5% and BY 72.5%). Only 44.4% sought out hospitals to obtain medical care. Only 35% of the subjects were able to show valid registration in the national health insurance scheme (46.3% EW and 27.5% BY). EW were asked about the use of PPE (Table 2). Overall, 25.3% reported that they wore some kind of protective equipment, of whom 57.1% used safety boots and 9.5% each wore safety glasses, dust masks, and helmets.

Regarding symptoms and diseases, there were no significant differences between the groups regarding infectious diseases, malaria, diabetes, gastrointestinal symptoms, high blood pressure or other cardiovascular symptoms, respiratory diseases, or psychiatric diseases (Table 3). High prevalences could be observed in both groups in reports of malaria in the last year (77.0%), digestive problems (62.9%), cough (64.2%), and various symptoms of mental problems (anxiety 45.2%, depression 34.5%, and posttraumatic stress disorder 46.9%). Significant differences between EW and BY for red itchy eyes (EW 67.9% and BY 51.6%), back pain (EW 91.6% and BY 79.6%), and work-related injuries (EW 75.0% and BY 42.6%) were observed (Table 3). No significant differences could be detected for skin diseases, shortness of breath, eye injuries, or hearing loss (Table 3). Half of the workers suspected an association between the symptom shortness of breathing and their occupation. For skin diseases, 41.7% of the e-waste workers but only 20.7% of the BY suggested an association between symptom and occupation (Table 3).

When relating certain symptoms and diseases with specific job tasks, we only observed a significant correlation between skin disease and dermal contact to chemicals or metals for all workers (*p <* 0.05). No significant associations could be observed for any of the following combinations: hard physical work and back pain or numbness of extremities, exposure to noise and hearing loss, and inhalation of smoke and shortness of breath (Table 4).

The distribution of injury location is shown in Figure 2, with hand injuries accounting for most injuries, followed by arms and legs. A significant correlation between work-related injuries and drug use among all subjects could be determined (Table 4).

Back pain was distributed differently among the different occupational groups, as depicted in Figure 3. Here, the prevalence was particularly high for the EW, with 100% of collectors and 92% of dismantlers complaining about back pain.

## 4. Discussion

To the best of our knowledge, this study was the first to comprehensively investigate work-related diseases in a study population of workers in Accra’s informal e-waste sector. This study compared symptoms and diseases with a group of non-e-waste workers, working at the Agbogbloshie recycling site without being involved in e-waste processing.

Both the e-waste workers and the control group reported many similar health problems, such as malaria, digestive problems, or mental disorders. The report of similar health problems is not completely unexpected. Both groups had a similar age distribution. The overall number of participating women was small, although there was a significantly higher number of females in the control group. Both groups had a similar smoking behavior and a similar conduct in seeking medical care. Both groups were exposed to *Anopheles spp.* during their work (Agbogbloshie is directly located next to a lagoon), which explains the malaria infections. Similarities in dietary habits could explain the common digestive problems. Finally, the high prevalence of mental problems among all participants can be explained by the migration history of most workers from both groups. Having migrated from poverty in remote areas (from northern Ghana as well as from neighboring countries) the mainly young men enter a challenging environment full of stressors and insecurities in Agbogbloshie. Experiences of violence and discrimination might increase their vulnerability to psychiatric disorders.

However, red itchy eyes, back pain, and work-related injuries had a statistically significant higher frequency in the group of e-waste workers. Red itchy eyes among e-waste workers can most likely be related to the direct exposure to fumes and smoke for the burners and irritating chemicals by manual dismantling. Another common symptom of e-waste workers was back pain, whether caused by lifting heavy objects or sitting in a bent position, while the electronic part is locked in place with the feet and broken open with a hammer and chisel. The forced working posture of dismantlers, as well as the lifting and carrying of heavy objects by onion carriers and sellers, might be responsible for severe musculoskeletal symptoms. Therefore, back pain was also frequently reported in the control group. The high consumption of drugs (e.g., cannabis), in addition to the use of painkillers, could be a way of dealing with the chronic pain and the intolerable working conditions. The high frequency of work-related injuries can be explained by specific work tasks, such as the manual dismantling. The workers suffer cuts, mainly on the upper extremities, from the pointed tools used for stripping the equipment. Gloves are rarely worn. An important co-factor concerning injuries could be the significantly higher substance abuse among e-waste workers. As demonstrated, an association between drug use and the likelihood of injury could be observed.

These results can be integrated into the existing literature. Burns et al. and Yu et al. assessed injuries among the e-waste workers at Agbogbloshie, although Burns et al. focused on the role of noise exposure and Yu et al. predominantly explored the health knowledge of e-waste workers [16,17,19]. Ohajinwa et al. observed that cuts on hands/fingers among e-waste workers were the most common injuries [18]. Here, a connection with the lack of PPE and the job designation was suspected, which we also observed. Similar occupational injuries were found among workers with solid waste in Addis Ababa, Ethiopia [20]. Among workers in the recycling sector in Santo André, Brazil, Gutberlet et al. not only noticed cuts and fractures, but the workers also described body pain, which was linked to ergometric causes [21], as seen in our study.

Besides occupational related factors, environmental conditions play a major role in Agbogbloshie as well. Being one of the world’s most toxic places [3], the severe pollution of water, air, and soil has to be taken into account while assessing adverse health effects. In 2013, Norman et al. reviewed studies on the role of environmental factors on health. Chemicals from the environment in air, water, and food are associated with an increased probability of a wide variety of non-communicable diseases [22]. A noxious exposure especially in early childhood development is described as being clearly associated with the pathogenesis of cancer [23], asthma [24], neurodevelopmental conditions [25], obesity [26], and chronic diseases [27,28]. Regarding the specific contamination by electronic waste, Grant et al. compiled all known health implications in a comprehensive review in 2013 [14]. Restricted thyroid function [29], cellular expression and function [30], adverse neonatal outcomes [31,32], changes in temperament and behavior [33,34], and decreased lung function [35] were detectable. There was a clear association with abortions [31], stillbirth, reduced birthweights [32], and DNA damage [36]. The extent of environmental contamination by informal e-waste recycling was investigated in various other studies. Caravanos et al. detected traces of aluminum, copper, iron, lead, and zinc in air samples and elevated levels of lead in soil samples at Agbogbloshie [7]. Oteng-Ababio et al. showed elevated PBDE levels in ashes, soils, and vegetables from Agbogbloshie [8]. Otsuka et al. found increased trace metal values in the earth of the Agbogbloshie market [37]. A study conducted by Greenpeace describes chemical contamination of soil on the site and in the sediment of the lagoon [9]. The results of a study by Hosoda et al. suggested a PCB (polychlorinated biphenyls) load of the Ghanaian coast by the e-waste site in Agbogbloshie [38]. In Guiyu’s electric scrap heap in China, the world’s largest, contamination of the earth with PAH and the riverine environment with heavy metals was attributed to e-waste related activities [39,40]. Sepulveda et al. summarized that very high levels of Pb, PBDEs, PCDD/Fs (polychlorinated dibenzodioxins/difurans), and PBDD/Fs (polybrominated dibenzodioxins/difurans) in air, bottom ash, dust, soil, water, and sediments were found in e-waste sites in China and India [41].

Based on this, Robinson already concluded in 2009 that local contamination at the site of e-waste activities, where e-waste workers are exposed to toxins through skin contact and inhalation, would spread into groundwater, air, and food chains of the environment [42]. Alabi et al. carried out a comparative cross-sectional study in Lagos, Nigeria, in 2015, which dealt with the public health effects on workers and residents. On the Alaba International Market and the Computer Village Market, the two largest markets for electrical goods in Nigeria, e-waste is either burned or simply disposed of in the market area. All workers and residents reported changes in the smell and taste of drinking water and health problems (aches, migraine, nausea, spontaneous abortions, and cancer) that were significantly different from an unexposed control group [43]. It is therefore not surprising that the bystanders also show symptoms that could be caused by the toxic substances in their environment.

Our study has several limitations. First, we could not apply a validated screening tool to systematically assess work related and non-work-related symptoms and diseases. In the absence of such a tool, we used focused questions based on ICD-10 (International Statistical Classification of Diseases and Related Health Problems) specifications after extensive consultation with occupational physicians and local project support. However, clear clinical diagnoses, physical measurements, and laboratory testing rather than self-reported complaints would have allowed more valid conclusions. As described, the study population was very heterogeneous in terms of regional origin and education. The questions therefore had to be translated into different languages with the help of interpreters, which may have affected their consistency. In addition, many subjects had low health literacy. This explains why not all answers could always be evaluated. The recall bias is also inherent in the system. Not all past complaints could be reliably described by the subjects. In addition, the healthy worker effect could have caused a bias towards a healthier study population, if sick workers did not show up at work and therefore did not appear in the study.

The results of our study provide concrete indications on how to improve the situation of workers by occupational health interventions. Although our study could only be a momentary snapshot, our findings should lead to clear actions. As Yu et al. already stated, many of the e-waste workers do not have sufficient knowledge about the health risks of their work [19]. Occupational safety training and equipping workers with adequate tools and personal protection equipment is therefore urgently needed. At the same time, our results clearly show the need for local medical care. Further research should focus on the long-term damage of informal e-waste recycling but should not lose sight of the existing needs.

## 5. Conclusions

The majority of the reported health problems did not differ between the e-waste workers and the control group of workers without involvement in e-waste processing. However, red itchy eyes, back pain, and work-related injuries were more frequent in the e-waste workers group. The occurrence of red itchy eyes can be explained by the exposure to eye irritating substances, such as fumes or chemicals during burning or manual dismantling. Back pain is a typical symptom for working in forced positions or by carrying heavy loads. Manual dismantling using inadequate work tools elevates the risk for work-related injuries. All of these adverse health conditions can be reduced by occupational safety training and the use of personal protection equipment. Since workers at the Agbogbloshie recycling site who were not involved into e-waste processing also frequently reported adverse health effects, these particular groups of workers should also get into the focus of occupational health research. An urgent need for occupational health interventions, adequate medical care, and more in-depth future research is clearly evident.

## Figures and Tables

**Figure 1 ijerph-17-01534-f001:**
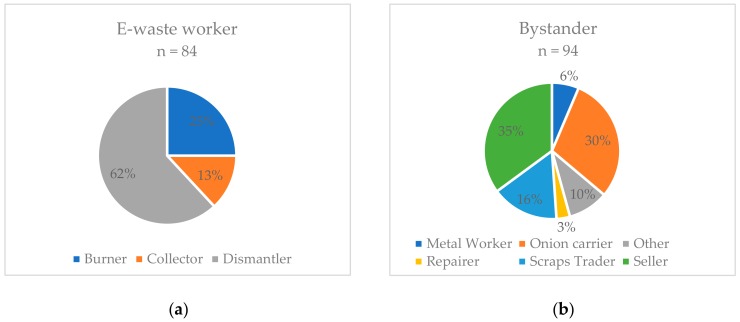
Job tasks of e-waste workers (**a**) and bystanders (**b**) (in percent of subgroups, total *n* = 178).

**Figure 2 ijerph-17-01534-f002:**
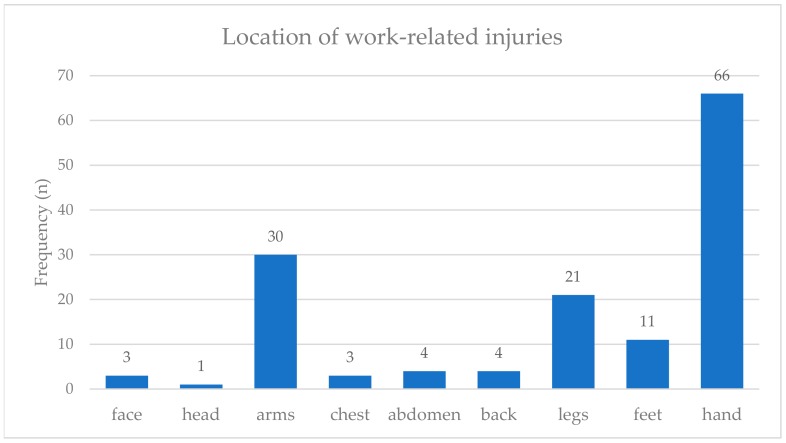
Distribution of the location of work-related injuries for all e-waste workers and bystanders at the scrap yard (*n* = 178).

**Figure 3 ijerph-17-01534-f003:**
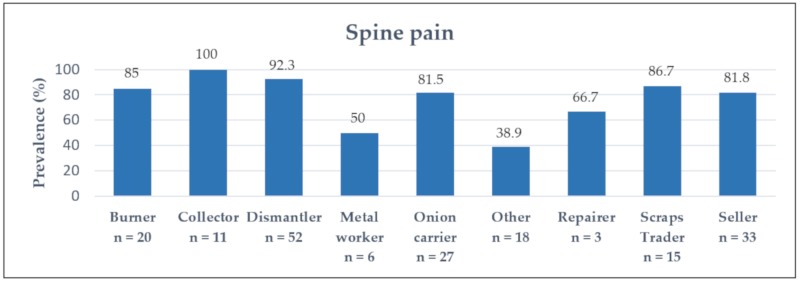
Frequency of back pain for all occupational groups (total number of workers *n* = 178).

**Table 1 ijerph-17-01534-t001:** Selected demographic characteristics of E-waste workers and bystanders at Agbogbloshie scrap yard.

Demographic Characteristics		All (*n* = 178)	E-Waste Workers (*n* = 84)	Bystanders (*n* = 94)	*p*-Value (Chi-Square)
Age (years)	mean	28	27	28	ns
Sex	male	149 (83.7)	82 (97.6)	67 (71.3)	<0.05
Marital status *n* (%)	Divorced	5 (2.8)	3 (3.6)	2 (2.1)	ns
	Married	93 (52.3)	44 (52.4)	49 (52.1)	
	Single	80 (44.9)	37 (44.1)	43 (45.7)	
Education *n* (%)		*n* = 177	*n* = 84	*n* = 93	<0.05
	None	62 (35.0)	22 (26.2)	40 (43.0)	
	Primary	38 (21.5)	20 (23.8)	18 (19.4)	
	Junior High	47 (26.6)	30 (35.7)	17 (18.3)	
	Senior High	26 (14.7)	11 (13.1)	15 (16.1)	
	Tertiary	4 (2.3)	1 (1.2)	3 (3.2)	
Origin *n* (%)	Ghana	140 (78.7)	84 (100.0)	56 (59.6)	<0.05
	Other countries	38 (21.4)	0 (0.0)	38 (40.4)	

ns = not significant.

**Table 2 ijerph-17-01534-t002:** Selected health characteristics of the e-waste workers and bystanders on the Agbogbloshie scrap yard.

Health Characteristics		All (*n* = 178, *n* (%))	EW (*n* = 84, *n* (%))	BY (*n* = 94, *n* (%))	*p*-Value (Chi-Square)
Smoking		46 (25.8)	19 (22.6)	27 (28.7)	ns
Drug use		27 (15.2)	21 (25.0)	6 (6.4)	<0.05
Medication		100 (57.5) *n* = 174	46 (57.5) *n* = 80	54 (57.5) *n* = 94	ns
	pain	80 (79.2)	34 (73.9)	46 (83.6)	ns
Access to medical care via	Hospital	76 (44.4) *n* = 171	34 (42.5) *n* = 80	42 (46.2) *n* = 91	
	Pharmacy	132 (77.2)	66 (82.5)	66 (72.5)	
	Traditional Healer	62 (36.3)	37 (46.3)	25 (27.5)	
National health insurance		56 (35.0) *n* = 160	25 (33.8) *n* = 74	31 (36.1) *n* = 86	ns
Use of PPE (only e-waste workers)			21 (25.3) *n* = 83		
	safety glasses		2 (9.5)		
	safety boots		12 (57.1)		
	dust masks		2 (9.5)		
	helmet		2 (9.5)		

ns = not significant.

**Table 3 ijerph-17-01534-t003:** Selected symptoms and diseases of the e-waste workers and bystanders on the Agbogbloshie scrap yard.

Symptoms and Diseases		All *n* = 178 *n* (%)	EW *n* = 84 *n* (%)	BY *n* = 94 *n* (%)	*p*-Value (Chi-Square)
Infections		10 (5.6)	4 (4.8)	6 (6.4)	ns
	Tuberculosis	7 (3.9)	2 (2.4)	5 (5.3)	
Malaria last 12 months		134 (77.0) *n* = 174	66 (79.5) *n* = 83	68 (74.7) *n* = 91	ns
Diabetes		2 (1.1)	1 (1.2)	1 (1.1)	ns
Digestive problems		112 (62.9)	57 (67.9)	55 (58.5)	ns
Hypertension		28 (15.8) *n* = 177	15 (17.9) *n* = 84	13 (14.0) *n* = 93	ns
Other cardiac symptoms		52 (29.6) *n* = 176	22 (26.5) *n* = 83	30 (32.3) *n* = 93	ns
	Palpitations	28 (15.9)	16 (19.3)	12 (12.9)	
	Chest pain	21 (11.9)	5 (6.0)	15 (16.1)	
Cough		113 (64.2) *n* = 176	53 (63.9) *n* = 83	60 (64.5) *n* = 93	ns
Mental disorders	Symptoms of anxiety	68 (38.4) *n* = 177	38 (45.2) *n* = 84	30 (32.3) *n* = 93	ns
	Symptoms of depression	61 (34.5)	30 (35.7)	31 (33.3)	ns
	Symptoms of posttraumatic stress disorder	83 (46.9)	44 (52.4)	39 (41.9)	ns
Skin symptoms		65 (36.7) *n* = 177	36 (43.4) *n* = 83	29 (30.9) *n* = 94	ns
	itching	55 (31.1)	33 (39.8)	22 (23.4)	
	light sensitivity	6 (3.4)	1 (1.2)	5 (5.3)	
	pain	13 (7.4)	3 (3.6)	10 (10.6)	
	caused by work ^1^	21 (32.3)	15 (41.7)	6 (20.7)	
Shortness of breath		49 (27.8) *n* = 176	25 (30.1) *n* = 83	24 (25.8) *n* = 93	ns
	caused by work ^1^	24 (49.0)	12 (48.0)	12 (50.0)	
Red itchy eyes		105 (59.3) *n* = 177	57 (67.9) *n* = 84	48 (51.6) *n* = 93	<0.05
Eye injuries		42 (23.7) *n* = 177	24 (28.6) *n* = 84	18 (19.4) *n* = 93	ns
Hearing loss		28 (15.8) *n* = 177	14 (16.7) *n* = 84	14 (15.1) *n* = 93	ns
Back pain		150 (85.2) *n* = 176	76 (91.6) *n* = 83	74 (79.6) *n* = 93	<0.05
	neck	77 (43.8)	37 (44.6)	40 (43.0)	
	back	138 (78.4)	73 (88.0)	65 (69.9)	
Work-related injuries		103 (57.9)	63 (75.0)	40 (42.6)	<0.05
	cuts	95 (53.4)	60 (71.4)	35 (37.2)	
	burns	35 (19.7)	23 (27.4)	12 (12.8)	

^1^ Workers were asked if they suspect an association between the symptom and their occupation. ns = not significant.

**Table 4 ijerph-17-01534-t004:** Association between specific occupational exposures and medical conditions (for all workers, *n* = 178).

	Occupational Exposure		Sum *n* (%)	*p*-Value (Chi-Square)
	Yes *n* (%)	No *n* (%)		
	Dermal contact to chemicals or metals			
Skin symptoms	44 (43.1)	20 (27.0)	64 (36.0)	<0.05
	Hard physical work			
Back pain	136 (85.0)	14 (87.5)	150 (84.3)	ns
Numbness	67 (49.2)	5 (35.7)	72 (40.4)	ns
	Volume exposure			
Hearing loss	22 (15.6)	6 (16.7)	28 (15.7)	ns
	Inhalation of smoke			
Shortness of breath	45 (28.5)	4 (23.5)	49 (27.5)	ns
	**Drug Use**			
	**Yes (%)**	**No (%)**		
Work-related injuries	21 (77.8)	82 (54.3)	103 (57.9)	<0.05

ns = not significant.

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
