# Peer review of "Health Consequences for E-Waste Workers and Bystanders—A Comparative Cross-Sectional Study"

_ijerph, 2020, doi:10.3390/ijerph17051534_

Round 1

Reviewer 1 Report

This article focuses on health outcomes among electronic waste recyclers in Ghana by comparing their health status against a group of bystanders. This research moves the e-waste literature past exposures and towards health outcomes, which is important in understanding the true occupational health impacts of e-waste recycling. The authors find that recyclers had increased complaints of "red itchy eyes" and back pain. Workers had a majority of injuries on their hands, and workers who were in direct contact with chemicals had increased reports of skin disease. It is suggested that this research can be used to help inform e-waste workers of their health concerns.

Broad Comments

Overall - It is a little confusing about what the point of the comparison group is. They have similar exposures, some more so than others. Are you trying to show the exact health conditions that are unique to e-waste work versus any exposure to e-waste?

Introduction - More information on exposures of the other occupational groups would be helpful for comparison purposes.

Methods - How did you define work-related versus non work-related disease? PPE use seems like a work-related disease, and some infections, digestive problems, cough, hypertension, and mental disorders can be work-related.

Discussion - Overall, there are some interesting findings here. However, the authors need to refocus the discussion portion to highlight these. The discussion goes through an in-depth literature review on several points, but does not sufficiently relate the findings to this paper. In some places, the discussion seems unrelated. Instead, the authors should refocus on discussing the pertinent findings and provide possible explanations for the findings. Your discussion should also talk about why you saw the differences between the e-waste workers and the bystanders - what is different?

Conclusion - The conclusion does not summarize the findings and instead focuses on possible interventions, which had not been mentioned prior to the discussion.

Specific Comments

Page 2 Line 40 - Were the participants recruited at the medical unit seeking medical care? If so, you should mention this in the limitations as your sample is biased.

Section 2.2 - In your limitations, it appears you have some sort of validated questionnaire you used. If so, you need to mention that here.

Table 1 - Did you investigate to see if there were differences by sex that might explain differences in the bystander population? 

Table 2 - What is PTBS? I don't see where it is defined.

Table 3 - Is there more specific information for the health characteristics? Itching is not necessarily a skin disease  (irritation, for example), but it can be (psoriasis, for example). This term is very broad. Same for the "pain" descriptor. 

Page 7, line 2 - I am not sure why this paragraph starts with moreover. If this was connected to the previous paragraph, I don't see the point that the "moreover" is contributing to.

Page 7, line 10 - It is not fair to say that this is the first comprehensive study (for example, see Seith et al, 2019), however it may be the first for the e-waste site in Accra.

Page 8, line 5 - These findings should be ignored. You included them in the study, and they are very interesting. Also, it is not clear what is meant by a "lack of resilience" but that is a sensitive argument to make without proof.

Page 8, line 11 - This is a large paragraph, and the point is unclear until the final sentence. It would be helpful to the reader to preface this information with your objective of the paragraph upfront.

Page 8, line 38 - It is not clear what the objective of this paragraph is. It begins by discussing other occupational injury findings, and end by talking about environmental issues from a public health stand point.

Reviewer 2 Report

Summary

The authors present a cross sectional survey of e-waste workers and a comparison group of non-e-waste workers in Agbogbloshie, Ghana. This study is important because it describes the health of e-waste workers in a country where we know little about working conditions and their impact on health. Many low and middle income countries are bearing the burden of global increases in e-waste and it is important that we better describe and quantify, where possible, the effects of this industry.

Comments

The questionnaire is not clearly described in the methods section. It is not clear what questions were asked or how diagnoses were determined. The reader needs more information on the questionnaire, the questions asked, and the interview process in order to interpret the results. (e.g., when and where were the questionnaires completed? Who asked the questions? Were validated questionnaire items used? etc.)  

It is also not clear how symptoms and diseases were determined to be work-related. Were participants asked explicitly about whether a health effect was work-related (e.g., worse at work or better away from work?) or did the authors make the determination? For example, cough (Table 1) could be work-related but is shown with the non-occupational symptoms and diseases – why? The same is true for mental health, infections and even cardiovascular. I suggest the authors give more thought to the organization of Table 1 and 2, and whether the symptoms/diseases need to be split into two tables.

In a cross-sectional study the healthy worker (HW) effect may impact study results. Both the HW selection effect and the HW survivor effect may impact who was captured in this study. The authors have not mentioned this bias, or how it may have affected this study. The HW effect and its possible impact should be included in the discussion section.

It would appear from Table 4 that the participants were asked about exposures at work. This was not mentioned in the methods. Why? This information should be included in a more detailed description of the questionnaire and interview process. All exposures included in the questionnaires should be noted.

Page 8, Line 5 – ignoring the mental health problems seems like a poor choice. Could these workers not be better supported? Just because it’s not caused by work, doesn’t mean it won’t impact their work or their health.

Overall, there are not many differences between the e-waste workers and the bystanders in terms of health. I would have liked to see more discussion of this. Why might this be? Were the bystanders recruited from the same physical location as the e-waste workers? Is it possible they also have similar exposures to the e-waste workers? If they do, this might not be a good comparison group. The comparison group should be similar in terms of demographics but be un-exposed to the hazards of interest.

Did the authors look at the relationship between drug use and workplace injury? This is a growing area of research interest around the world.

Comparisons between e-waste workers and bystanders (Table 1) would benefit from statistical testing and presentation of the resulting p-values.

Figure 2 and 3 are missing y-axis labels. The numbers in the figures also need to be more fully described. Particularly Figure 3. The tables and figures should stand alone so that the reader does not need to look to the text to understand what is being presented.

Page 2, Line 41 – please spell out the acronym RWTH

Table 4, in the sum column please also show the % so that the reader can quickly determine the prevalence of exposure.

Round 2

Reviewer 2 Report

The authors have submitted a much-improved manuscript.

A few additional comments:

In the discussion the authors reference the Grant (2013) review, which is itself an important paper.  But. it would be preferable to also cite the primary studies for the outcomes mentioned in lines 18-23 (page 9)

Figure 3 is a bit confusing to me. What is the main message the authors want to convey? If it is that the prevalence of back pain differs between the groups, I would suggest showing prevalence (%) on the y-axis and the groups on the x-axis, and adding the sample size (n) for each group into the group label. This would make the bars immediately comparable. Right now the bars are more reflective of the group size, rather than the prevalence of back pain.

The formatting for Table 2 needs to be revised; it is very hard to read. I will defer to the editor, but I would think it acceptable to drop the sample size (n) values in the table and add a footnote noting that the exact sample size differs slightly by question due to data/questionnaire completeness. This is a common problem in survey studies, especially when the participants complete the survey themselves.

In table 4, I believe the percentages for drug use are not in alignment with the other numbers in the table. For consistency you should be showing the column percentages, not the row percentages:

Drug use = yes; work related injuries 21/27 (78%)

Drug use = no; work related injuries 82/151 (54%)
